# Mechanical and Chemical Resistance of UV Coating Systems Prepared under Industrial Conditions Using LED Radiation

**DOI:** 10.3390/polym15234550

**Published:** 2023-11-27

**Authors:** Milena Henke, Barbara Lis, Tomasz Krystofiak

**Affiliations:** Department of Wood Science and Thermal Techniques, Faculty of Forestry and Wood Technology, Poznan University of Life Sciences, Wojska Polskiego 28, 60-627 Poznan, Poland; milena.henke@up.poznan.pl

**Keywords:** UV lacquer system, LED radiators, honeycomb sandwich board, high-density fiberboard, nanoindentation, scratching, impact, abrasion, cold liquid, roughness

## Abstract

The furniture industry constantly strives to search for ecological and cost-effective solutions in the production of wood-based composites. It is anticipated that furniture with a honeycomb core and HDF-facing will gain market share. Understanding how specific technical and procedural factors on the finishing line affect the resistance of coatings on furniture elements made of honeycomb boards was the main goal of the study. With the use of a digital microscope, the roughness of two different types of HDF was tested. On the industrial UV LED+Hg finishing line, 198 different surface coating variations were produced by applying five or six layers of varnish applied, ranging from 3 to 30 g/m^2^ and hardening them with various surface power densities. On the basis of statistical tests, the influence of individual factors on abrasion, impact, and scratch resistance was determined. The nanointendence test of the coatings was used to measure the hardness and elasticity modulus. The coloring caused by coffee traces was checked using a colorimeter. The findings confirm the conception that LED+Hg lamp modules can replace mercury and gallium-doped mercury lamps.

## 1. Introduction

The production of wood-based boards increased by 109% between 2009 and 2019 [1]. They are becoming more and more popular due to their easier processing and lower price than solid products [2]. Wood-based panels joined together to create wood-based composites. They combine aesthetic and economic values while maintaining appropriate strength and deformation characteristics [3,4]. Such systems include a honeycomb composite panel, which consists of a particleboard or MDF (medium-density fiberboard) frame filled with honeycomb paper, covered on both sides with thin fiberboard or plywood facing. The inspiration of the hexagonal shape taken from nature allows for high durability while effectively filling the space with the least possible material consumption [5,6,7,8]. This structure makes furniture and doors lightweight, which is helpful for transportation, handling, and assembly [9]. Because of the growing demand for cost-effective materials, the use of honeycomb paper in the furniture industry is expected to grow. These materials are becoming increasingly popular in the design of tables and other furniture due to their high strength-to-thickness, strength-to-weight ratio, and acoustic properties [10]. The market value of honeycomb panels is anticipated to rise 6.5% from 2023 to 2033, reaching around USD 2.26 trillion, according to the most recent industry research by Fact. MR [11]. The vast bulk of the rapidly expanding Asian furniture market’s products are composites with honeycomb cores [12].

The popularity of low-cost, light-weight boards with honeycomb cores must be accompanied by high-quality furniture finishing. Most consumers base their furniture decisions on aesthetic considerations [13]. HDF (high-density fiberboard) is most often used as a facing for boards with a honeycomb core. Due to the speed of cross-linking and the mechanical and physical properties of the coatings, manufacturers are increasingly using UV-cured varnishes to finish such furniture [14,15]. Applying one layer of coating to achieve optimal resistance and aesthetics for the product would be extremely difficult. The majority of multi-layer coating systems for organic coatings that have been studied in research are just two or three layers thick [16,17,18,19]. Multi-layer coatings are used in industrial surface finishing to meet a variety of customer requirements. Wood finishing systems consist of stain, basecoat, primer, and topcoat. Due to the requirements that each layer must meet, every varnish has a different set of characteristics. The problem is that single layers do not behave like multilayer coatings [20,21,22]. Mechanical resistance tests concentrate primarily on the topcoat layers, which bear the load [20,23,24]. It has been shown that while elastic deformation can spread all the way to the substrate, plastic deformation of the system may only reach the initial layers [19,25,26]. The physical properties of coatings are designed with specific applications in mind and require a balance between flexibility and coating hardness [22]. According to the investigations, the physical characteristics of the product in the liquid state, the functional characteristics of individual layers, and the type of substrate also have an effect on the overall strength of the coating system facing as stress propagates into deeper layers [27,28,29,30].

Due to new laws and environmental standards, UV-cured coatings must continually advance [31]. Mercury lamps are the basis of UV curing technology. The emission of UV (visible and infrared radiation) results from the evaporation of mercury into the plasma at high temperatures [32]. The challenge with UV energy is the depth of cure. The energy decreases as the varnish layer’s thickness rises [33]. To prevent that, gallium-doped mercury lamps are used for thick, pigmented coatings [30]. In 2017, the international treaty of the Minamata Convention obliged 137 member countries to limit the usage of mercury [34,35]. The European Commission’s RoHS2 directive indicates the possibility of using mercury in the production of mercury lamps for UV curing by 2027. There are many regulations, including REACH and the TSCA/Lautenberg Act, as well as regulatory authorities, e.g., the United States Environmental Protection Agency and the United Nations Environment, that aim to create consumer awareness so that their demands accelerate changes in the industry [36,37,38]. An alternative to high-energy mercury lamps is LED diodes. They can be used for radiation curing through radical polymerization, which is an evolving sector in the coatings industry [39]. UV LEDs use 50–75% less energy than equivalent mercury lamps. They do not emit radiation of different wavelengths and do not generate high temperatures, which makes them safer for the cured material. They can be turned on and off immediately. Mercury lamps are ready for operation just a few minutes after being turned on. Additionally, harmful mercury is eliminated, and ozone formation is reduced. All of these factors combined make it a green technology that can greatly lower UV curing technology’s carbon impact [40,41,42,43]. Research by Allied Market Research showed that the global UV LED market is expected to reach USD 1.2 billion by 2026, up from USD 271.1 million in 2018, representing an annual growth rate of 17.3% from 2018 to 2026 [44].

This technology faces several challenges. It has been established that there is not enough current chemistry to properly cure UV LED sources at the required line speeds. The matching of photoinitiators, curing of topcoat layers, machine settings of LED lamps, and the speed of painting lines are still being researched [42,45,46,47]. There are few reports on the comparison of the resistance assessment of coatings cured with LED lamps and traditional mercury lamps. Coatings cured solely with LED lamps will achieve worse results than coatings cured with mercury lamps. Curing the coatings with mercury-LED hybrid modules allowed for comparable or better results for both systems [30,42].

Despite the growing pressure to introduce UV curing technology using LED lamps, there is still a lack of clear information related to the process of curing coatings using LED technology. This poses a challenge for researchers to support the industry in changing coating curing technologies. The aim of this study was to examine the impact of selected variables on the mechanical properties and chemical resistance of coatings. These include the density of the fiberboard (HDF), sanding parameters, the amount of basecoat and topcoat applied, the number of layers, and various surface power densities on an industrial line combined with LED and mercury lamps. The tests were carried out at a conveyor belt speed of 50 m/min. It was assumed that these factors may have a significant impact on the resistance of the coating, and knowing them and determining their relationships may support entrepreneurs and production technologists and become a starting point for further work on varnish formulations and LED lamps.

## 2. Materials and Methods

### 2.1. Materials

A board on a frame with a honeycomb core was used for the tests. The frame was made of particleboard, and its core was filled with recycled paper with a honeycomb structure. HDF boards were used as facing (Figure 1).

The pressed elements on an industrial multi-shelf press had dimensions of 700 mm × 390 mm × 22 mm; the pressing pressure was 1.2 kg/cm^2^ and took 170 s. The frame was made of particleboard with a thickness of 17 mm and a density of 540 ± 20 kg/m^3^ (Kronospan, Szczecinek, Poland). With a 1.45 g/m^2^ honeycomb as the core of the board, the compressive strength was 2.5 kPa at 23 ± 2 °C and 50 ± 5% humidity (Axxor, Zwolle, The Netherlands). Two types of HDF with a thickness of 2.5 mm were selected. The results of the tests carried out on the physical and mechanical properties of HDF are presented in Table 1.

The chosen boards were joined using a one-component PVAC adhesive with a viscosity of 14.000 mPa.s measured with a METLER TOLEDO S210 meter at 40 ± 0.5 °C and a pH of 4.1. The units made in this manner with a total thickness of 22 mm were permitted to start the finishing process after the 96 h conditioning period (temperature 23 ± 2 °C and humidity 50 ± 5%). A multi-layer painting technique was adopted, using putty, basecoat, and topcoat (Table 1). The paints utilized are industry-standard products. The manufacturer’s specified paint composition is as follows—heavy putty: calcium carbonate ≥ 40–<60%; ethoxylated propylidynotrimethanol, esters with acrylic acid ≥ 20–<40%; 4,4′-Isopropylidenediphenol, oligomeric reaction products with 1-chloro-2,3-epoxypropane, esters with acrylic acid ≥ 5–<10%; and 1-metoksypropan-2-ol ≤ 0.5%. Basecoat: titanium dioxide ≥ 40–<60%, propylidinetrimethanol ≥ 10–<20%; 4,4′-Isopropylidenediphenol, oligomeric reaction products with 1-chloro-2,3-epoxypropane, esters with acrylic acid ≥ 5–<10%; alkoxylated pentaerythritol tetraacrylate ≥ 5–<10%; diacrylic dipropylene glycol ≥ 3–<5%; polymer based on polyoles and modified acrylic acid esters ≥ 2.5–<5%; phenyl-bis(2,4,6-trimethylbenzoyl-)phosphine oxide ≥ 0.1–<1%; and 2-propenoic acid, 1,1′-[(1-methyl-1,2-ethanediyl) bis[oxy(methyl-2,1-ethanediyl)]]ester, reaction products with diethylamine ≥ 0.1–≤0.5%. Topcoat: propylidinetrimethanol, ethoxylated, esters with acrylic acid 20–40%; 4,4′-Isopropylidenediphenol, oligomeric reaction products with 1-chloro-2,3-epoxypropane, esters with acrylic acid 10–20%; titanium dioxide (1-methyl ethane-1,2-diyl)-bis[oxy(methyl ethyl-2,1-diyl)] diacrylate 10–20%; diacrylate (1-methyl ethane-1,2-diyl)-bis[oxy(methyl ethyl-2,1-diyl)] 5–10%; 2,2-bis(acryloxymethyl)butyl acrylate 2.5–5%; methyl benzoylformate 2.5–5%; amorphous silica (silica gel, precipitated silica) 1–2.5%; tetrabutylammonium bromide 1–2.5%; 2-propenoic acid, 1,1′-[(1-methyl-1,2-ethanediyl) bis[oxy(methyl-2,1-ethanediyl)]]ester, reaction products with diethylamine ≤ 0.5%; and photoinitiator ≤ 0.1%.

### 2.2. Surface Lacquer Finishing Process

For testing, 198 different samples with a white finish were prepared. The research selected 7 factors for which two or three values were tested. Two alternative methods of grinding HDF boards were used (I: 180–220, 220–220; II: 220–220, 400–400), and sandwich boards were pressed using two types of HDF with different densities (850 kg/m^3^ and 830 kg/m^3^). Three basecoat application values (40, 45, and 50 g/m^2^) and three topcoat application values (3, 6.5, and 10 g/m^2^) were selected. Another variable was the use of a different number of varnish applicators (five or six), choosing the distribution of the total amount of applied basecoat on two or three applicators. Curing with mercury lamps was used in three surface power density variants: 120, 90, and 60 W/cm^2^, similarly in the case of LED lamps 12, 9, and 6 W/cm^2^. The materials and finishing process were prepared under technological conditions on a varnish line (Borne Furniture, Gorzów Wielkopolski, Poland). The UV source in the research was a mercury lamp and a 395 nm LED. The lamps were placed 10 mm above the surface of the sample.

The UV roller coating line where the experiments were performed is shown in Figure 2. Sanding the components was the first step in the process. The Heesemann LSM8 + EA10 wide belt sanders were used for the tests. Then, for each version, two layers of putty were applied in a constant combination of 30 and 15 g/m^2^. With the help of three mercury lamps (Hg), the putty was cured. The following process involved grinding. Then, apply the basecoat. Two LED lamps were used to harden the first layer of basecoat. For each of the tested versions, a second layer was applied in a fixed quantity (30 g/m^2^). Two LED lights and one mercury lamp were used to cross-link the second layer of the basecoat. Some coats had three layers of basecoat. Two LED lamps were used for hardening. Five mercury lamps and two LED lamps were used to harden the final topcoat layer. At a speed of 50 m/min, a temperature of 27 °C, and a relative humidity of 41%, all of these procedures were carried out on a single production line. Figure 2 displays the process diagram. This work is a continuation of previous research on the mechanical and chemical resistance of coatings cured with mercury lamps [48].

### 2.3. Nanointendance Research

The study of mechanical properties at the nanoscale was performed using the Nanovea PB1000 Nanointender. For the hardness and modulus of elasticity tests, a Berkovich diamond cone indenter was used for the study. The indenter moved toward the contact load value of 0.08 mN at a speed of 5 m/min to start the test. The load was then gradually increased to 200 mN and held there for 10 s before the indenter returned to the initial position. The data were transformed using a computer program, and the results of hardness and modulus of elasticity were then displayed. For the study, twelve representative samples were chosen and cut to 50 × 50 mm sizes. Six tests were run for each variant [49,50,51,52].

### 2.4. Determination of Scratch Resistance

A Clemen tester, described in PN-88 F-06100/11, was used to determine the scratch resistance. For 198 variants, three samples with dimensions of 100 mm × 100 mm were prepared. The specimens were mounted on a table, and a knife with a blade protruding 0.8 mm was set on the surface. The table and the board were moved manually at a speed of approximately 15 mm/s. Three measurements were taken on each sample [53].

### 2.5. Impact Test (Ball Method)

Impact resistance was assessed based on the assumptions specified in the ISO 4211-4 standard. The testing tool consists of a steel tube with specific heights marked (25, 50, 100, 150, and 200 mm). A cylindrical weight falls from a certain height and hits a steel ball. The ball is located at the bottom of the test tube and directly contacts the coating of the tested sample, transferring the impact energy to it. The test was performed by dropping a weight onto a ball from all heights (15, 25, 50, 100, and 200 mm). The study was performed for all 198 variants. Five tests were performed for each height per sample. The degree of damage to the test area was assessed by reference to a descriptive numeric rating code (Table 2). Using a Brinell magnifier, the largest diameter of the impact trace was measured in each test area [54].

### 2.6. Abrasion Resistance

The standard ISO 7784-3 was used to assess the abrasion resistance of the surface finishes. The test was performed on a Taber Abraser 5130. Before starting the test, a 100 mm × 100 mm sample was drilled at the central point, and the edges were secured with tape. Finally, the specimen was weighed. Then, it is mounted on a spindle placed on the table. Two wheels covered with abrasive material were pressed against the board using weights. CS-10 abrasive wheels were used. The table with the sample rotated around a vertical position. The surface of the specimen was abraded with the wheels. The abrasive effect was the result of contact between the sample and the wheels. Each board was weighed after 100 and 200 cycles. The weight loss was determined based on the difference between the weight before and after each test cycle. The abrasion test was performed on three samples for 198 variants [55].

### 2.7. Resistance to Cold Liquids

Surface resistance to cold liquids was determined according to the standard EN 12720+A1. The test was performed on two boards for 198 variants of surface finish. Cold liquids that are typical of everyday situations in households were selected for the test (Table 3).

Filter paper discs were immersed in the test liquids for 30 s before application with tweezers. The test surface was immediately protected against evaporation with an inverted Petri dish. The test was performed for two time intervals: 16 h and 24 h. After the appropriate time interval, the Petri dish and the disc were removed with tweezers. Any residual liquid was removed with absorbent paper. After 24 h, the surface was wiped with a cloth soaked in a cleaning solution and then washed with water. The test surface was rated according to Table 4 [56].

For selected samples in which the test trace was observed, the color was measured. Twin samples were checked for an applied color test. Color was evaluated according to the CIELAB system (EN ISO 11664-4, 2011) using a comparison colorimeter (DT-145) with an illumination/geometry angle of 45°/0°. The measuring aperture was 8 mm, and the light source (illuminant) was D65. For each sample, 10 measurements were taken, and average values were recorded. Color differences were determined using the delta E parameter, which indicates the total color difference [57,58,59].

### 2.8. Roughness Measurement

Surface topography was analyzed using a non-invasive method using a Keyence VHX 7000 digital microscope with a 3D topography module. For this purpose, two types of samples with different density HDF facings were selected from among the tested variants: board type 1 with a density of 850 kg/m^3^ and board type 2 with a density of 830 kg/m^3^. The digital microscope was calibrated according to the manufacturer’s recommendations. The magnification was set to two hundred times, and the brightness and contrast of the image were adjusted. Surface roughness tests were performed: arithmetic mean deviation (Sa), geometric average roughness (Sq), and ten-point height (Sz) on the 2 cm^2^ of the sample surface. After the study was completed, numerical reports of roughness parameters were generated.

### 2.9. Data Processing

Microsoft Excel 2019 and Minitab 19 software programs were used for the statistical analysis of the test results. In order to determine the influence of individual factors on the resistance of the surface, the normality of the distribution was checked, and an analysis of variance was performed. The main effects chart was used to present the data. A Pareto effect chart was drawn to show the effect sizes. A box plot, interval plot, fitted line plot, interaction plot, point chart, and bar chart were generated to show the obtained results.

## 3. Results

### 3.1. Nanointendance Research

The distribution’s normality was verified using the Ryan–Joiner test, which is analogous to the well-known Shapiro–Wilk test. Hardness and elasticity modulus were checked. The gathered measurement data had a normal distribution for raw data at the significance level of 5%. At the predetermined confidence level (>0.05), the null hypothesis—that is, that the variance of the dependent variable error was the same in every group—was accepted [60].

For the gathered data, the average hardness main effect screener was created (Figure 3). Important factors that influence hardness can be identified using the Pareto chart of the effect size in standard deviations. This test focuses on assessing which factors have the greatest impact on the variables under study and which are less important. It was determined that the following five are the most influential factors (with effects larger than two): The surface power density of UV Hg took first place. Then, the results of sandpaper, basecoat, and the surface power density of LED were comparable. The last factor from the group with the greatest influence turned out to be the amount of topcoat. Then, the number of applicators belonged to the next group of effects (between 1 and 2). The type of board was the factor that marginally affected the test’s outcome.

In the case of testing the modulus of elasticity, three factors had the greatest impact: the surface power density of UV Hg, the surface power density of LED, and the amount of basecoat (larger than 2). The factor in the second group was the amount of topcoat. The rest had a marginal impact on the test results.

Figure 4 shows the influence of individual factors on the values of the hardness and elasticity modulus parameters. The HDF had no impact on the obtained hardness result or elasticity modulus. The higher grit of the sandpaper provided 23% higher hardness results. The outcome was the highest for all tested systems.

The best hardness result was obtained for coatings hardened with mercury lamps with a surface power density of 90 and 120 W/cm^2^ and with LED surface power density of 9–12 W/cm^2^. In the case of mercury lamps, a decrease in surface power density to 60 W/cm^2^ was associated with a 38.4% decrease in hardness, and in the case of LED lamps, a decrease from 9 to 6 W/cm^2^ was associated with a 28% decrease in hardness. The lowest modulus of elasticity was recorded at the lowest surface power density of mercury and LEDs, and the highest at the middle surface power density.

The hardness and modulus of elasticity of the coating decreased as the amount of basecoat applied increased. The greatest reduction was between 45 and 50 g/m^2^ and amounted to 17.5 ± 0.5% for both factors. Increasing the amount of topcoat applied resulted in a hardness increase in several percentages. The graph depicting the effect of topcoat amount on modulus of elasticity differs. The increase from 3 to 6.5 g/m^2^ is associated with a 14.0% increase in the modulus of elasticity. An increase from 6.5 to 10 g/m^2^ results in a 10.5% decrease in modulus. Increasing the number of layers results in an increase in the tested parameters by several percentages.

It was decided to check the dependence between the modulus of elasticity and the hardness of the coatings. Figure 5 shows the nonlinear fitting model. The standard error of the estimate (S), which tells us how scattered the data are, is 0.140259. The model has a good fit with the data. The coefficient of determination R-Sq (77.2%) and the adjusted coefficient of determination R-Sq (adj) (72.2%) suggest that the model is stable and well fits the equation [61]:Elastic Modulus [GPa] = −0.658 + 29.85 Hardness [GPa] − 86.55 Hardness [GPa]^2^


The hardness is the resistance of the coating to penetration and deformation under the action of a standardized indenter [25]. Schuh drew attention to the tip of the indenter, which is never atomically sharp and shows significant blunting, which may affect the obtained results [62]. Pavlič investigated waterborne finish, one-component solvent-borne finish, and tung oil on invasive wood species. As a result, hardness was associated with the interaction between the substrate and the coating [63]. It has been shown that elastic deformation can reach the substrate [26,64,65]. Schwalm proved that highly cross-linked coatings are hard and brittle below the polymer’s glass transition temperature (Tg) and soft and flexible above Tg. These characteristics are combined in water-based varnish coatings cured with UV radiation [66]. Nail gel coatings must achieve the highest modulus of elasticity and hardness at the same time [67].

Hermann examined the hardness of UV-cured coatings. Cross-linking density (CLD) has a direct effect on hardness, as formulations with the highest CLD values had the highest hardness [19]. It was found that the hardness of UV-cured epoxy acrylate coatings containing nanoparticles is lower than the hardness of samples without nanoparticles [68]. On the other hand, the hardness and nanoindentation modulus of hybrid silicones and hybrid epoxy coatings increase with the addition of nanofillers [69]. Borysiuk et al. applied coatings of acrylic varnish to the surfaces of beech and pine wood. A light filler of 20 to 25 g/m^2^, two primers of 15 g/m^2^, and a topcoat of 5 to 6 g/m^2^ were all applied. There were two ways to harden the coatings. The first variant was hardened with mercury–gallium lamps, and the second with LED lamps using mixed LED+Hg modules after the first layer of primer and topcoat. The Buchholz approach was used to compare the hardness of the coatings. He did not notice any significant differences in both coating finishes; the only factor influencing the result was wood species (beech and pine) [30]. Landry et al. evaluated water–acrylate solutions on glass plates and sugar maple (Acer saccharum March) samples. All of the coatings were 4 mm thick. After initial drying, the samples were hardened using an LED lamp at a radiation intensity of 467 mJ/cm^2^ and a conveyor speed of 0.8 m/min. The UV Hg formulations were hardened at an irradiance of 456 mJ/cm^2^, and the conveyor belt speed was 11 m/min. The Konig pendulum was used for research on the hardness of coatings cured with UV-Hg and UV LED lamps on glass plates to ensure substrate uniformity. The coating that was cured with a traditional mercury lamp indicated greater resistance. The use of UV-LED lamps may result in insufficient polymerization of the preparation [42].

### 3.2. Scratch Resistance

The Ryan–Joiner test, which is comparable to the well-known Shapiro–Wilk test, was used to confirm the normality of the distribution. The gathered scratch resistance measurement data had a normal distribution for raw data at the significance level of 5%, and the null hypothesis regarding the equality of error variances in all groups was accepted at the specified confidence level (>0.05) [60]. Data were assessed using ANOVA and seven different variables: the different surface power density of the LED and UV Hg, the amount of basecoat and topcoat applied, the number of applicators used, the different sandpaper on the grinder, and the type of HDF (Table 5). The average scratch resistance values were discovered to differ at the α = 0.05 significance level and to be statistically significantly different for the amount of basecoat, amount of topcoat, UV-Hg, and LED surface power density (*p*-value < 0.05). There was no statistically significant difference in the number of applicators, the type of board, or the grinding sandpaper utilized (*p*-value > 0.05).

A main effect plot (Figure 6) was created to assess the impact of individual factors on scratching. The highest scratch resistance of 1912 N was achieved by applying a basecoat of 40 g/m^2^. The resistance decreased by 23 N when the varnish application was increased to 45 g/m^2^. A further increase to 50 g/m^2^ resulted in a decrease of 107 N. Increasing the number of basecoat applicators had better results of 34 N. Applying 3 and 10 g/m^2^ of topcoat gave results that differed by only 10 N, while applying varnish in the amount of 6.5 g/m^2^ gave the weakest results, lower by 50–60 N. It is important to note that the group of results with the application of 6.5 g/m^2^ of topcoat had the lowest standard deviation. The surface curing had the greatest impact on scratching. The surface cured with less energy generated worse results. A difference of 132 N was measured between the surface power density of UV Hg, 60 W/cm^2^ and 120 W/cm^2^. This value was 86 N for extreme LED surface power density. However, there is a noticeable stabilization of this parameter above a surface power density of 9 W/cm^2^. The scratch resistance results were unaffected by the board used or the method of grinding.

The general full factorial method was adapted to the additional analysis of scratch resistance. To show which combinations of components are important, the Pareto graph of standardized effects was presented (Figure 7). At the 0.05 level, all influencing factors that were higher than the baseline (2.131) were significant. The significance of four combinations was determined in this manner: the amount of basecoat and the surface power density of LED; the number of applicators and the surface power density of LED; the number of applicators and the surface power density of UV Hg; the type of board; and the amount of topcoat.

Figure 8 depicts interaction charts between factors that show the correlations between them. When the amount of basecoat was increased for the variants cured with LED lamps with a surface power density of 6 W/cm^2^, a significant reduction in scratch resistance was noticed. Scratch resistance was comparable for basecoat amounts of 40 and 45 g/m^2^ cured with a higher surface power density of LED (9 and 12 W/cm^2^). Only the increase to 50 g/m^2^ resulted in a scratch resistance decrease.

The following dependencies are related to the number of basecoat applicators used. The use of six layers (three basecoats included) resulted in the same scratch resistance for different surface power densities. Using only five layers (including two basecoats), with the same total weight of basecoat applied, the resistance when hardening with 6 W/cm^2^ lamps was 260 N lower than with 12 W/cm^2^ lamps.

Scratch resistance for UV Hg improves with higher surface power density. In the case of UV Hg with a surface power density of 90 and 120 W/cm^2^, scratch resistance is higher when six applicators are used, while in the case of lamps with a surface power density of 60 W/cm^2^, it remains unchanged.

The last statistically significant interaction occurs between the type of HDF and the amount of topcoat. Scratch resistance increases with the increase in density of the HDF board if the topcoat layer is between 6.5 and 10 g/m^2^. When applying 3 g/m^2^, the resistance decreases as the density of the board increases.

Pavlič proved that an additional layer of topcoat improved coating resistance [29]. The use of additional layers under the topcoat and increasing the thickness of the coatings improves scratch resistance [70,71,72]. Scratch tests indicated that the substrate (various types of wood-based boards) and the type of finish used (epoxy, polyester, and epoxy-polyester powder paint and water-based liquid paint) had an impact on the scratch results [73]. Other researchers have indicated improved scratch resistance using the addition of cellulose to waterborne varnish and tris(2-hydroxyethyl) isocya-nuran-triacrylate in the case of UV acrylic varnish [74,75]. ZnO films deposited on a polyimide interlayer improve scratch resistance by 50% [76].

Borysiuk et al. showed that the method of hardening the coatings (UV and UV-LED) does not clearly affect scratch resistance. Coatings applied to the beech surface are twice as resistant as coatings applied to pine wood. The beech coating cured with LED lamps obtained a higher result than in the case of that cured with Hg-Ga lamps [30]. Landry et al. discovered no difference in scratch resistance between coatings hardened with Hg-UV and UV-LED lamps in scratch resistance tests [42]. The authors tested the scratch resistance of the same coating variants in a previous publication (only 48 variants resulted from a lack of differentiation in the surface power density of LED). They found that LED+Hg-cured UV coatings had significantly improved scratch resistance [48].

### 3.3. Impact Resistance

Based on the test performed, the marks were assessed in accordance with Table 3, and their diameter was measured. A total of 99% of variants achieved a rating of 3 for a fall of 200 mm (Figure 9); 64% of the marks were rated 3 after a fall of 100 mm, and 36% were rated 4. In the case of a fall from 50 mm, 97% of the samples were rated 4 and 3% were rated 3. A total of 95% of the marks were rated 4 after a fall from 25 mm; 100% of the marks after a fall from 15 mm were rated as 5. The average diameter for a fall from 200 mm is 5.93 mm; for a fall from 100 mm, it decreased by 18%; and for a fall from 50 mm, it decreased by 43%.

The normality of the distribution of the mark diameter was verified using the Ryan–Joiner test, similar to the popular Shapiro–Wilk test. At the significance level of α = 5%, the impact diameters collected from the impact resistance measurements were normally distributed for raw data. ANOVA was performed, and the data were assessed on the basis of seven variables: the different surface power density of the LED and UV Hg used to harden the surface of the coating, the amount of basecoat applied, the amount of topcoat applied, the number of applicators used, the different sandpaper on the grinder, and the type of HDF. Only one factor: type of HDF obtained a *p*-value < 0.05 (amounted to 0.000). The marks on the higher-density HDF board were 6.8% smaller for a 200 mm fall, 13.3% smaller for a 100 mm fall, and 11.0% smaller for a 50 mm fall (Figure 10).

Researchers have demonstrated that the topcoat layer is harder than the basecoat layer. It may be an indenter for basecoat layers. Wood is softer, porous, and heterogeneous, resulting in a higher error rate than glass or metal substrates. The elastic deformation spreads further through the multi-layer coating and into the wooden substrate [51,77,78,79]. Well-cured coatings were shown to be one of the most influential parameters for the understanding of indentation and wear resistance [19]. The authors tested the impact resistance of the same coating variants in a previous publication. Both LED + mercury lamps and UV mercury and gallium-doped mercury lamp hardening technologies have similar ratings and average impact diameters [48].

Pavlič used the ball method to test the impact resistance of water-based coatings on the surface of oak floorboards. When the ball was dropped from 50 mm, the coating cracked, and when an additional layer of varnish was applied, the coating cracked when the ball was dropped from 100 mm. He also discovered that a coating’s impact resistance is largely determined by interactions between the coating and the substrate [29,59]. Lis examined the UV resistance of acrylic coatings on MDF. No resistance was determined when the ball dropped from a height of 400 mm, and resistance was assessed at grade level 2 when dropped from a height of 200 mm [74]. Homańska-Kaniewska examined the properties of acrylic coatings on pine and beech wood. The addition of biopolymers increased the impact strength of the coating [80].

### 3.4. Abrasion Resistance

The Ryan–Joiner test, which is comparable to the well-known Shapiro–Wilk test, was used to confirm the normality of the distribution. The results gathered from abrasion resistance measurements were normally distributed for raw data at a significance level of 5%. ANOVA was performed. Seven variables were used to evaluate the data: the different surface power density of the LED and UV Hg used to harden the surface of the coating, the amount of basecoat and topcoat applied, the number of applicators used, the different sandpaper on the grinder, and the type of HDF. The average weight loss values were discovered to differ at the α = 0.05 significance level and to be statistically significantly different for type of board (after 100 cycles, *p*-value 0.000; after 200 cycles, *p*-value 0.000) and amount of basecoat (after 100 cycles, *p*-value 0.027; after 200 cycles, *p*-value 0.024).

The multi-variable interval chart depicts how statistically significant factors discovered influence mass loss (Figure 11). During the first 100 cycles, HDF type 2 lost 34.3% more weight than HDF type 1. After 200 cycles, the difference in weight loss was 30.1%. An increase in the amount of basecoat applied from 40 to 45 g/m^2^ resulted in an increase of 20.7% for board type 2 with a lower density and 17.7% for board type 1 with a higher density. Another increase in the amount of basecoat from 45 g/m^2^ to 50 g/m^2^ resulted in a decrease in weight loss of 14.4% for the HDF with a lower density and a decrease of 22.2% for HDF type 1.

The vertical lines indicate the confidence interval, taking into account the individual standard deviation. In the case of mass loss after 200 cycles, the width of the confidence interval is much larger, which means greater uncertainty in the mean estimate.

For the obtained data, the Pareto graph of standardized effects was determined (Figure 12). The interaction between the amount of topcoat and the type of board was found to be the most statistically significant. The same combination appears in the first place of the Pareto chart for weight loss after 100 and 200 cycles.

The relationship between the type of board and the amount of basecoat is shown in Figure 13. The highest mass loss was found for board type 2, which had a lower density. The weight loss was similar when topcoat 3 and 6.5 g/m^2^ were applied, but it dropped by 17% when topcoat 10 g/m^2^ was applied. In the case of a board with a higher density, the weight loss increased with the increase in the topcoat. The weight loss was two times smaller when applying 3 g/m^2^, compared to a board with a lower density. However, when applying 10 g/m^2^, it was only 12% lower.

The obtained results are consistent with the literature. Herman et al. indicated the importance of the basecoat in achieving the mechanical strength of the coatings [22]. Tests on oak wood indicated that the additional topcoat increased the number of cycles required to achieve the Initial Point of Wear by 28% [29]. Akkuş et al. investigated the abrasion resistance of coatings made from various systems, including water-based and powder paints, particleboard, MDF, and plywood substrates. It was discovered that a coating with greater thickness increased resistance. The substrate’s effect on coating resistance was also observed. The low MDF board results are due to the thin layer of varnish applied to the MDF board, which has a high surface density. The MDF board, on the other hand, has a higher abrasion resistance due to its porous surface [81]. The research and development of varnish formulations with the highest abrasion resistance is currently underway. It was discovered that silicon nanoparticles improve abrasion resistance [82]. Higher results can also be obtained by increasing the amount of photoinitiator. However, manufacturers’ purpose is to limit the amount of photoinitiator [83]. The presence of colors, metals, and ceramic fillers had no effect on the coating’s abrasion capabilities [84,85]. Tests on UV coatings cured with LED lamps revealed that highly functionalized monomers accelerate coating hardening and improve coating abrasion resistance [86].

Researchers achieved varied results when comparing the abrasion resistance of coatings cured using LED lamps and mercury lamps. The authors investigated the abrasion resistance of the same coating variants in a previous publication (the lower number of variants stemmed from the absence of distinction in the LED surface power density). The weight loss of coatings hardened with LED+Hg lamps was greater than that of coatings hardened with mercury lamps [48]. Landry et al. evaluated abrasion resistance using the Taber Abraser. The test showed that the weight loss of samples cured with a UV-LED lamp is 1.5 times greater than that of samples cured with a mercury lamp [42]. Borysiuk et al. checked the abrasion resistance of acrylic coatings hardened with mercury and gallium-doped mercury lamps and LED+Hg lamps. The hardened coating using traditional mercury–gallium lamps was characterized by a 25–40% lower abrasion resistance (depending on the substrate) [30].

### 3.5. Resistance to Cold Liquid

The surface of all samples received a grade of 5 in the 16 h test. There was no discoloration or change in gloss. In the 24 h test, there was a small coffee mark left on the surface, which was rated 4 (Figure 14). The surface power density of the UV was the most important factor in determining the outcome of the 24 h study. Most samples with a lower rating were cured with mercury lamps with a surface power density of 60 W/cm^2^. The remaining part was cured with 90 W/cm^2^ lamps. However, not all variants hardened with a surface power density of 60 W/cm^2^ showed surface marks (60%). The number of applicators used was the distinguishing feature of the defect-free samples. Increasing the number of applicators from 5 to 6 with the same amount of varnish applied resulted in a higher degree of curing and the lack of any marks. Another factor was the use of a combination of 60 W/cm^2^ mercury lamps and 12 W/cm^2^ LED lamps. Analyzing the variants with a rating of 4, which were cured with a surface power density of 90 W/cm^2^, it can be concluded that they were always combined with LED lamps with a surface power density of 6 W/cm^2^. As a separate factor, the impact of LED surface power density was insignificant.

It was decided to use a colorimeter to assess color changes to determine the coffee’s influence after 24 h. All damaged varieties received the same score (4) in the previous subjective method of visually measuring chemical resistance. The color difference between the test sample and the control sample is represented by the dE* parameter. According to the literature, a dE < 1 is not visible to the observer [87,88]. Figure 15 shows the main effect plot for the obtained dE* results. A difference in color of 2.14 was observed for the application of basecoat in the amounts of 40 and 45 g/m^2^. An increase in the amount of basecoat applied to 50 g/m^2^ resulted in an increase in the color difference by 73% to the level of 3.7. Board 1 with a higher density recorded a greater dE* difference of 0.8 than board type 2. The surface power density of LED had no significant impact on the dE* parameter. The smallest difference in color was recorded for the highest surface power density (12 W/cm^2^).

The can coating research confirmed that better hardening improves coating chemical resistance, which corresponds to the results obtained [89]. Borysiuk proved that curing the coating with LED or mercury lamps has no effect on the coating’s resistance [30]. Henke et al. examined the chemical resistance of the same coating variants in a previous publication, curing only with high-temperature Hg + Ga lamps. Traces of coffee, wine, and acetone were detected in the 24 h and 16 h test periods when curing with mercury lamps. Better results were obtained with the coating cured with LED+Hg lamp modules. In both cases, the lowest results were obtained for the lowest surface power density of UV Hg [48].

UV acrylic coatings applied via pneumatic spraying were characterized by high resistance to selected cold liquids, except ethanol. The use of different compositions of coating substances, such as tris(2-hydroxyethyl)isocyanurate triacrylate, did not affect the result [90]. Pavlič proved that there is no connection between resistance to cold liquids and the chemical properties of varnishes [29]. Tests of acrylic and water-based varnish on Black Alder wood (*Alnus glutinosa* L.) did not obtain the maximum score when using alcohol [91]. Kayn tested the resistance of polyurethane, polyester, cellulose, and synthetic varnish coatings on oriental beech (*Fagus orientalis* L.) wood. Depending on the varnish used, the element that left a mark on the coating after 16 or 24 h changed [92]. The WB renewable UV resin performs better than a traditional 1 k castor-based PUD by giving better chemical resistance. Coffee and mustard marks were found during the test [93]. Other research has been conducted to compare the properties of three WB UV coatings with commercially available solvent-based conversion varnish, water-based conversion varnish, and water-based pre-catalyzed lacquer. All WB UV coatings show excellent chemical resistance with the exception of resistance to vinegar, 50% ethanol, and mustard in the 24 h test [94]. Dual-cure coating technology improves the chemical resistance of coatings [95].

### 3.6. Roughness Parameter

Due to the high significance of the factor represented by the board in these studies, it was decided to examine the topography of the boards. The linear roughness profile was determined, and the surface roughness of HDF panels used as facings for the honeycomb panel was examined (Figure 16 and Figure 17). Arithmetic mean deviation roughness (Sa) and geometric average roughness (Sq) were similar for both boards and were Sa 6.15 µm and 6.26 µm, Sq 7.76 µm and 7.84 µm. The largest difference of 20% was observed for the ten-point height (Sz) parameter. For the higher density board, it was 80.5 µm, while for the lower density board, it was 64.83 µm.

Panels made of wood, particularly high-density fiberboard, are more homogeneous than wood [96]. Nevertheless, the literature research indicates that there is no general tendency regarding the relationship between density and wood-based panel roughness [97,98]. It is also important to consider research on how the processed species and level of chemical degradation affect the seasonal fluctuation of particleboard characteristics [99,100]. Boards constructed of poplar, birch, and scots pine have somewhat greater values of surface roughness than fibers created from a blend of scots pine and beech chips. The roughest surfaces are found in boards composed of beech and oak fibers [101,102]. Pressing parameters and fiber thermal treatment may have a significant impact on the surface roughness of fiberboards [103,104,105]. Furthermore, observations from the literature suggest that the density profile may be more important for shaping the surface topography than the average density [106,107].

## 4. Conclusions

According to nanointedance research, an increased amount of topcoat may increase hardness but not necessarily elastic modulus. Increasing the amount of basecoat applied leads to a decrease in the hardness and elastic modulus of the coatings. The type of HDF had no effect on the hardness or elastic modulus. However, increasing the grit of the sandpaper increased the hardness. A correlation was noticed between the modulus of elasticity and the hardness of the coating, which was described using a quadratic function.

According to the results of the scratch testing, a positive correlation between resistance and the surface power density (Hg and LED) was proven. Applying less basecoat varnish improves scratch resistance. A large impact of interactions between surface power density, the amount of varnish applied, the number of coating layers, and the type of HDF was demonstrated. The scratch resistance often improved as the number of layers increased.

The type of HDF board used was crucial in the testing for ball impact and abrasion resistance. The HDF with a larger density provided greater resistance. The type of board and the amount of topcoat interacted significantly in the abrasion test. For the greater density board, the weight loss with the smallest topcoat application was 103% less.

In the 24 h test to determine the resistance of the coatings to cold liquids, only coffee marks were found. The marks were discovered on samples that had been hardened with mercury lamps using a low surface power density and a lower number of layers. In the dE* colorimeter analysis of coffee marks, the effect of basecoat quantity was demonstrated. The highest value of the dE* parameter was obtained with the largest amount of basecoat.

On the basis of this work and the authors’ other publications, it is possible to assess how using LED+Hg radiation modules may affect coatings. Compared to coatings hardened using mercury and gallium-doped mercury lamps, the LED+Hg tests revealed that the coatings had enhanced chemical and scratch resistance. Coatings cured with LED+Hg lamps obtained lower abrasion resistance and comparable impact resistance results than coatings cured with mercury and gallium-doped mercury lamps.

## Figures and Tables

**Figure 1 polymers-15-04550-f001:**
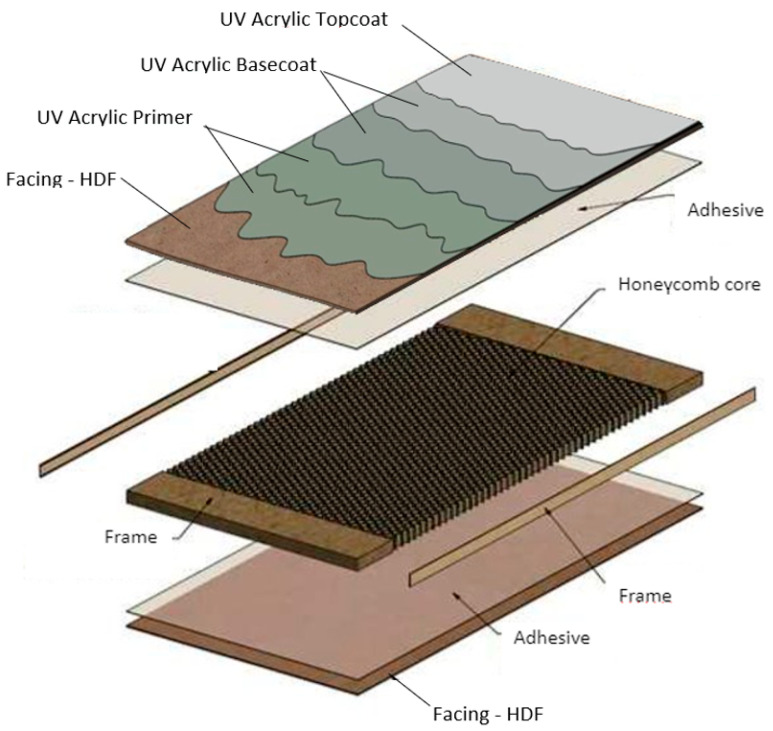
An overview of the sample structure [48].

**Figure 2 polymers-15-04550-f002:**
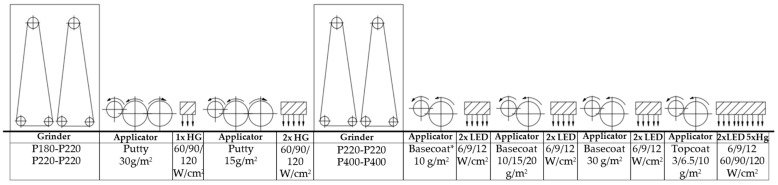
An overview of the varnish line on which the tests were conducted.

**Figure 3 polymers-15-04550-f003:**
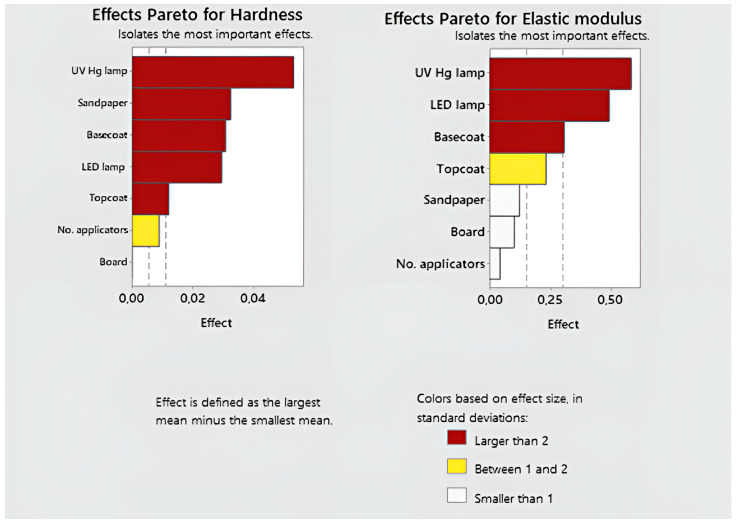
Pareto effects for hardness and elasticity modulus.

**Figure 4 polymers-15-04550-f004:**
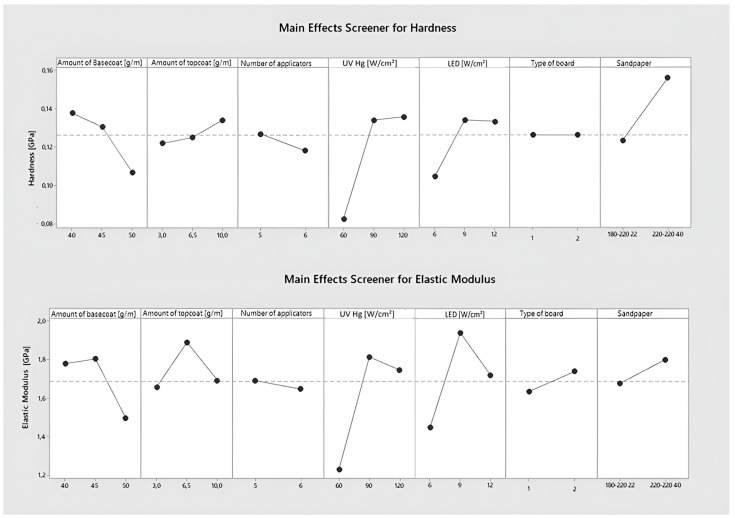
Main effect plot for hardness and elasticity modulus.

**Figure 5 polymers-15-04550-f005:**
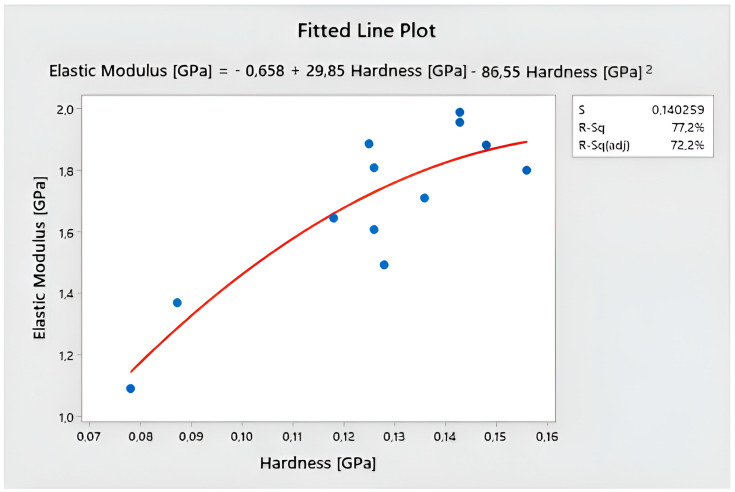
Fitted line plot and model for elastic modulus and hardness.

**Figure 6 polymers-15-04550-f006:**
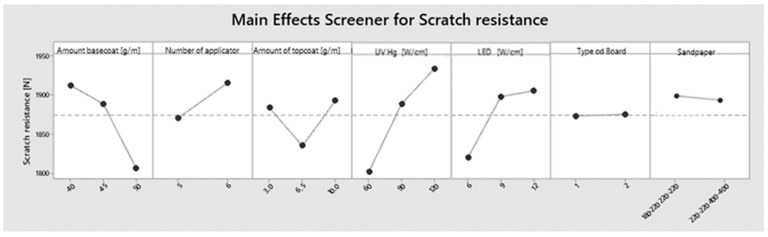
Main effect plot for scratch resistance.

**Figure 7 polymers-15-04550-f007:**
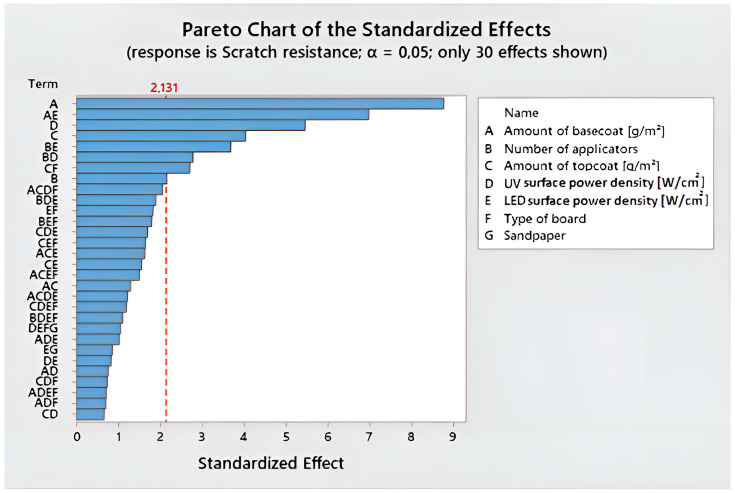
Pareto chart for scratch resistance.

**Figure 8 polymers-15-04550-f008:**
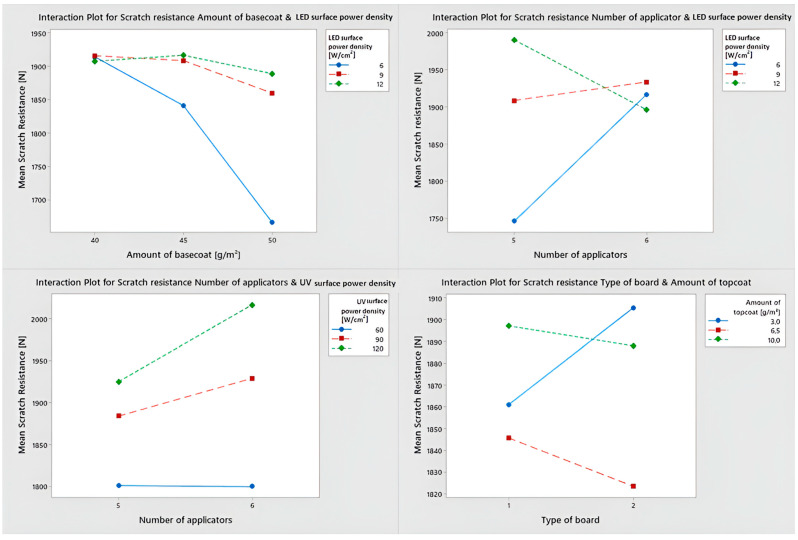
Interaction plot for statistically significant combinations.

**Figure 9 polymers-15-04550-f009:**
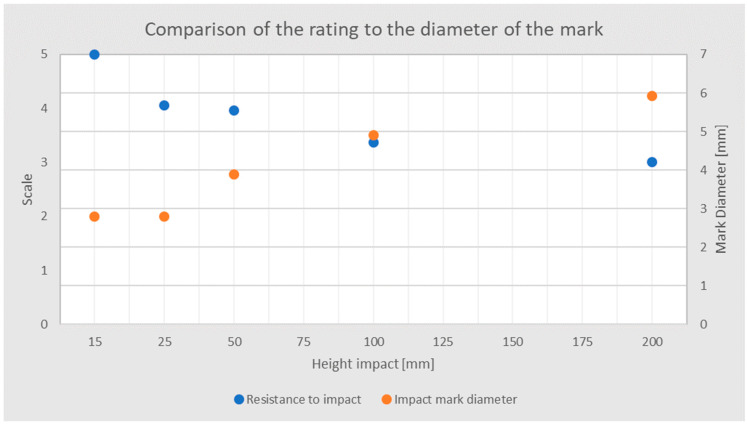
Comparison of the rating to the diameter of the mark.

**Figure 10 polymers-15-04550-f010:**
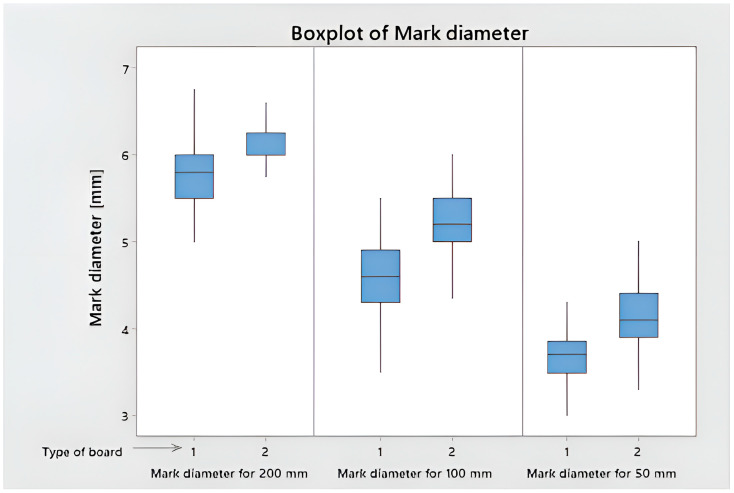
Box plot of mark diameter for drops from 200, 100, and 50 mm.

**Figure 11 polymers-15-04550-f011:**
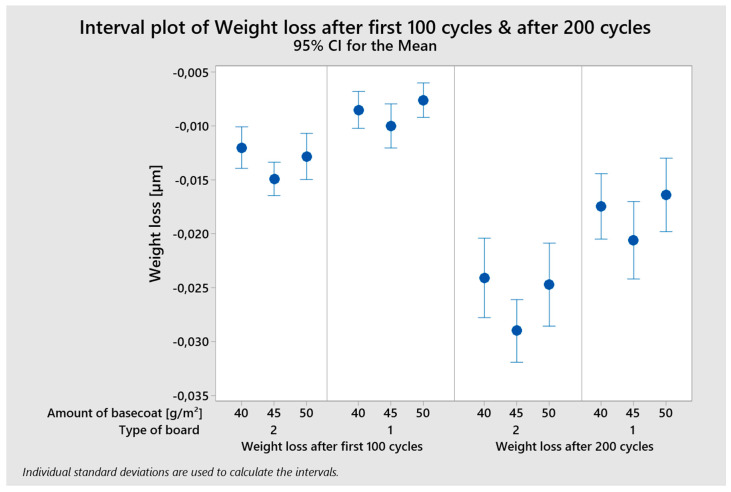
Multi-variable interval plot of weight loss.

**Figure 12 polymers-15-04550-f012:**
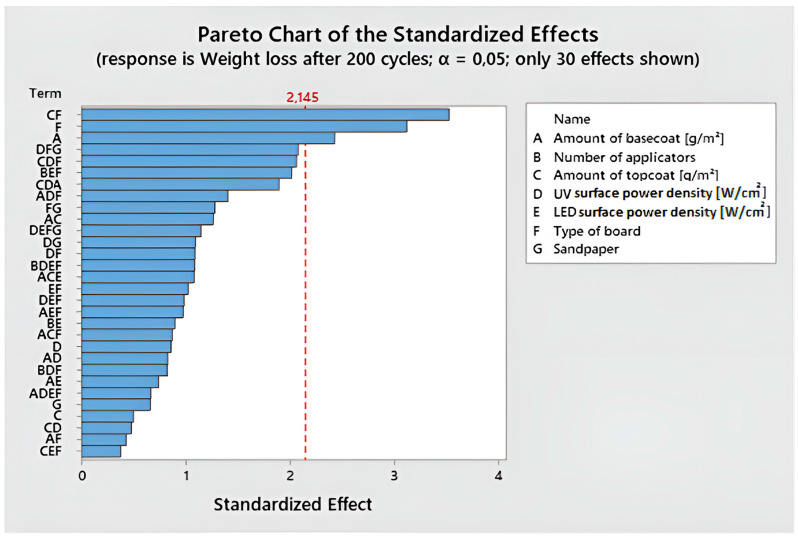
Pareto chart of the standardized effects for weight loss after 200 cycles.

**Figure 13 polymers-15-04550-f013:**
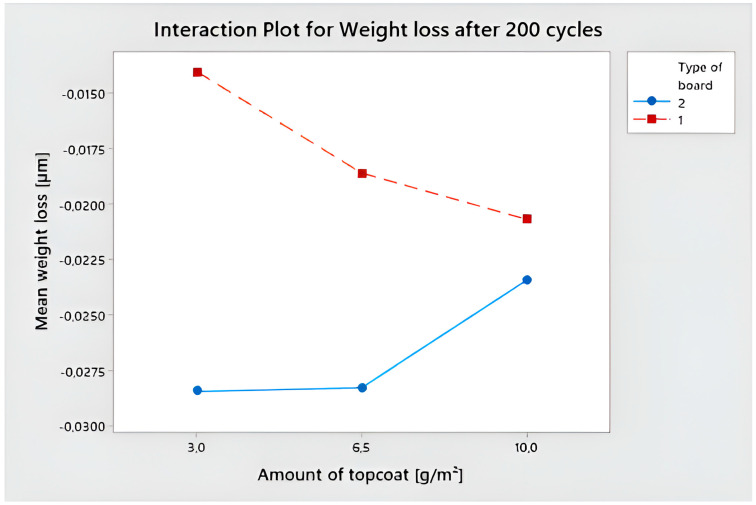
Interaction plot for statistically significant combination.

**Figure 14 polymers-15-04550-f014:**
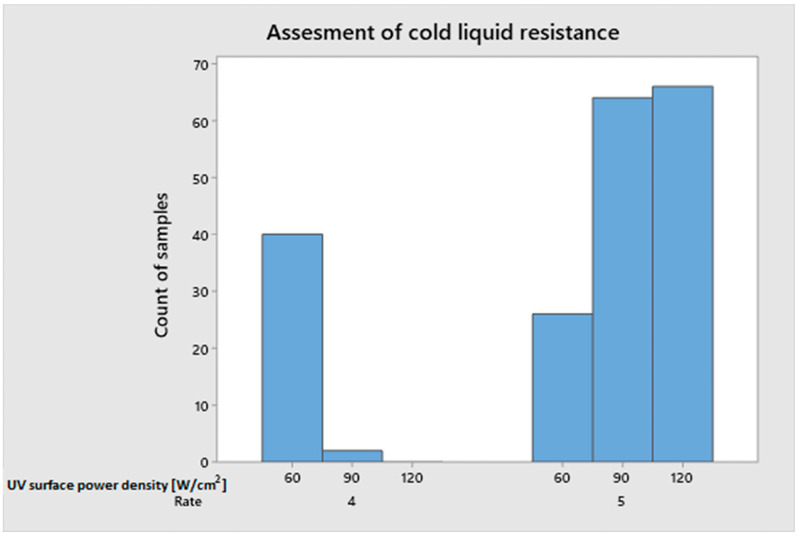
Assessment of resistance to cold liquids in terms of surface power density.

**Figure 15 polymers-15-04550-f015:**
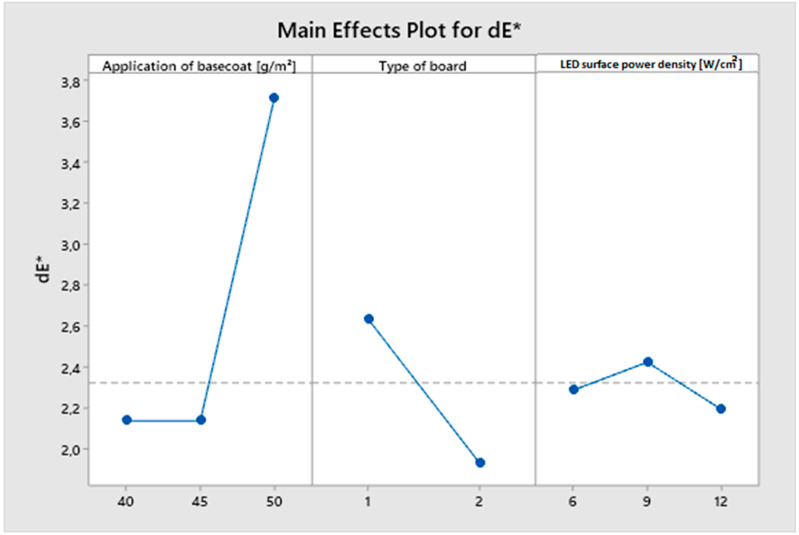
Main effect plot for dE*.

**Figure 16 polymers-15-04550-f016:**
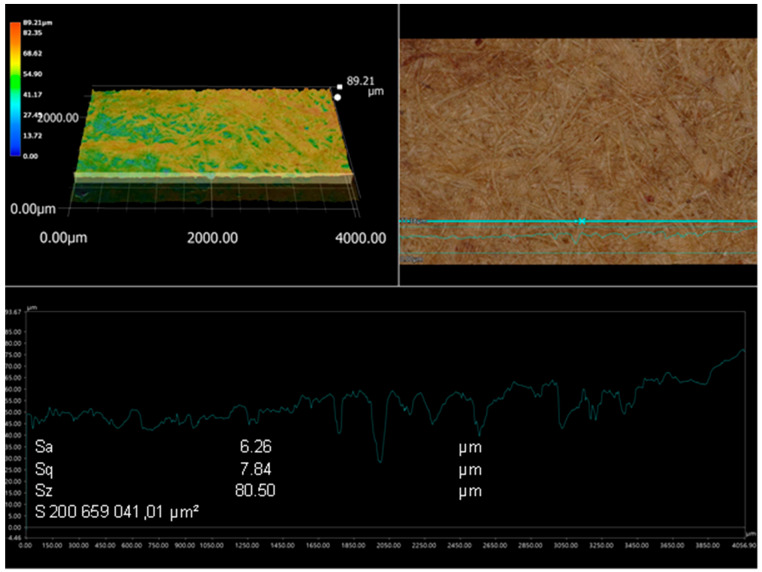
View of the surface topography of HDF type 1, density of 850 kg/m^3^.

**Figure 17 polymers-15-04550-f017:**
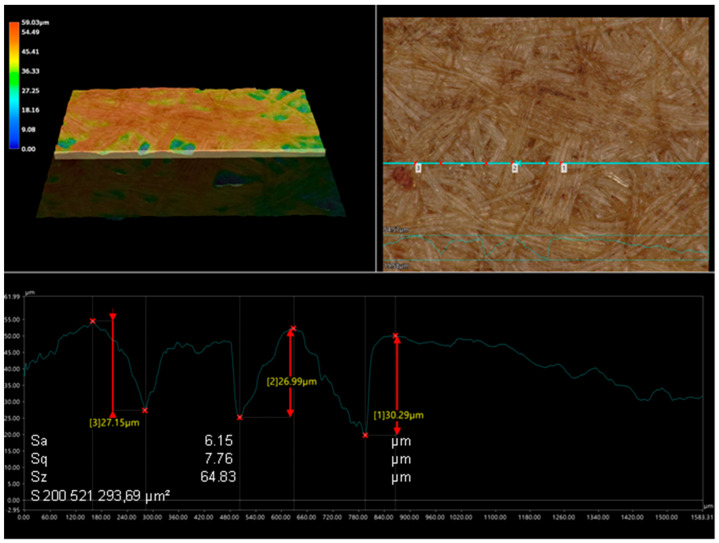
View of the surface topography HDF type 2, density of 830 kg/m^3^.

**Table 1 polymers-15-04550-t001:** Basic properties of the HDF and varnish products.

Parameter	HDF		Type of Varnish		
	A	B	Heavy Putty	Basecoat	Topcoat
Density (kg/m^3^) acc. to DIN EN 323:1993	850	830	-	-	-
Modulus of elasticity (MPa) acc. to the DIN EN 310:1993 standard	4.300	4.500	-	-	-
Humidity (%)	7	7	-	-	-
Swelling resistance (%) acc. to the DIN EN 317:1999 standard	45	45	-	-	-
Density (g/cm^3^)	-	-	1.63 ± 0.15	1.73 ± 0.15	1.30 ± 0.15
Solids content (%) acc. to the PN-EN ISO 3251:2019 standard	-	-	95.3 ± 0.5	98.3 ± 0.5	97.8 ± 0.5
Viscosity (mPa.s) (Brookfield, Thermosel 35 °C, 20 rpm, spindle 27)	-	-	7.700	400	1.475

**Table 2 polymers-15-04550-t002:** Ball impact evaluation criteria [54].

Rating	Criteria
5	No visible marks on the surface
4	No cracks on the surface, but an impact mark is visible only when the light from a light source is reflected off the test surface at, or quite close, to the test point back to the observer’s eyes
3	Slightly cracked surface, generally one or two circular cracks around the impact mark
2	Moderate-to-heavy crack formation within the limits of the impact mark
1	Crack formation beyond the impact mark and/or flaking of the surface finish or surface covering material

**Table 3 polymers-15-04550-t003:** Cold liquids used.

Cold Liquid	Characteristic
Distilled water	-
Acetone	-
Paraffin	Paraffinum liquidum
Ethylene	48% (*v*/*v*) aqueous solution
Wine	Merlot Trevenezie IGT 2021
Tea	1.75 g of tea leaves infused in 175 mL of boiling water, leached for 5 min without stirring, and then carefully decanted
Coffee	40 g of instant, freeze-dried coffee, dissolved in 1 L of boiling water
Beetroot juice	100% beetroot juice (Biurkom Flampol Sp. z o.o., Szczecinek, Poland)
Black currant juice	pasteurized nectar, black currant juice from concentrated juice (26%), fruit content minimum of 26%, (Tymbark-MWS Sp. z o.o., Tymbark, Poland)
Condensed milk	8% fat content, sweetened (Milk Company in Gostyn, Gostyn, Poland)

**Table 4 polymers-15-04550-t004:** Surface evaluation criteria [56].

Degree	Description
5	No visible changes (no damage)
4	Slight change in gloss—visible only in the reflection of a light source, e.g., discoloration or change in color or gloss; no change in the surface structure, e.g., swelling, fiber elevation, cracking, or blistering
3	Slight traces of damage (gloss)—visible from multiple perspectives, e.g., discoloration or change in color or gloss; no change in the surface structure, e.g., swelling, fiber elevation, cracking, or blistering
2	Strong traces of damage—visible in all viewing directions, e.g., discoloration, change in color or gloss, and/or the surface structure has changed slightly, e.g., swelling, fiber elevation, cracking, or blistering
1	Strong damage—the surface structure has changed noticeably, discoloration or change in color or gloss, the surface material has partially or completely come off, and/or the filter paper sticks to the surface

**Table 5 polymers-15-04550-t005:** Conducting a one-way ANOVA to assess how scratch resistance varies with the process variables.

One-Way ANOVA Response	Source	DF The Total Degrees of Freedom	Adj SS Adjusted Sums of Squares	Adj MS Adjusted Mean Squares	F-Value	*p*-Value
Scratch resistance	Surface power density of UV (W/cm^2^)	2	597,093	298,547	30.18	0.000
	Surface power density of LED (W/cm^2^)	2	293,629	146,814	12.81	0.000
	Amount of basecoat (g/m^2^)	2	376,220	188,110	17.05	0.000
	Amount of topcoat (g/m^2^)	2	119,508	59,754	4.84	0.009
	Number of applicators	1	33,406	33,406	2.62	0.107
	Sandpaper	1	29,039	29,039	2.28	0.133
	Type of HDF	1	154	154.1	0.01	0.913

## Data Availability

The data presented in this study are available on request from the corresponding author.

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
