# Peer review of "Mechanical and Chemical Resistance of UV Coating Systems Prepared under Industrial Conditions Using LED Radiation"

_polymers, 2023, doi:10.3390/polym15234550_

Round 1

Reviewer 1 Report

Comments and Suggestions for Authors

Wood-based composites are widely used in furniture industry, etc. However, curing coatings technology and its performance have a great influence on the furniture elements. Authors studied the mechanical and chemical resistance of UV coatings. It can be found that the conception of that Hg+LED lamp modules can replace Hg-Ga lamps. However, the current form of this study cannot be acceptable. Some aspects as listed below:

1. In Figure 2, more details about the process can be given? Besides, the image quality can be improved.

2. In Figure 3, how to define the influential factors of hardness and E modulus? More details can be given.

3. What is the function of the empirical model of Elastic modul and Hardness?

4. How to verify the coatings had enhanced chemical and abrasion resistance after curing of Hg+LED lamp? More details can be given.

Author Response

Dear Reviewer,

thank you very much for your valuable comments. Our answers are listed below and in the file.

yours sincerely,

Tomasz Krystofiak

Wood-based composites are widely used in furniture industry, etc. However, curing coatings technology and its performance have a great influence on the furniture elements. Authors studied the mechanical and chemical resistance of UV coatings. It can be found that the conception of that Hg+LED lamp modules can replace Hg-Ga lamps. However, the current form of this study cannot be acceptable. Some aspects as listed below:

Answer:

Thank you very much for your valuable comments.

  1. In Figure 2, more details about the process can be given? Besides, the image quality can be improved.

Answer:

New graphics have been drawn with more and higher quality details.

  1. In Figure 3, how to define the influential factors of hardness and E modulus? More details can be given.

Answer:

The Pareto test with the effect size was utilized to demonstrate the relationship between the applied parameters and their impact on the outcome of the production process, specifically on hardness and elasticity modulus in this case. It enabled the assessment of which factors had the greatest influence on the variables under investigation and which were less significant. The test considered factors such as the amount of applied lacquer, surface power density, the number of applied coatings, and the type of board. Subsequently, the effect size was calculated for each factor, determining the degree of its impact on the outcome variable. The obtained effect size value indicated the significance of the influence of a given factor, allowing for their prioritization from the most to the least influential. This resulted in the Pareto chart, which classifies factors based on their impact on the outcome variable. It indicates which factors were most significant (e.g., those with a large effect size) and which were less important. Based on this, three significance groups were identified (larger than 2, between 2 and 1, less than 1).

  1. What is the function of the empirical model of Elastic modul and Hardness?

Answer:

Correlations between the results of the investigated parameters were sought. In the case of the elasticity modulus and hardness, a function was successfully obtained with the level described by the formula. A quadratic function was fitted with the formula:

Elastic modulus [GPa] = -0.658 + 29.85 Hardness [GPa] – 86.55 Hardness [GPa]^2, exhibiting high model fitting parameters with S=0.140259, R-Sq=77.2%, R-Sq(adj)=72.2%.

Additionally, a linear function was fitted (Figure 1), Elastic modulus = 0.4551 + 0.1034 Hardness [GPa], but it yielded inferior model fitting parameters with S=0.142929, R-Sq=73.7%, R-Sq(adj)=71.1%. Therefore, the decision was made to present the quadratic function in the article.

Figure 1

  1. How to verify the coatings had enhanced chemical and abrasion resistance after curing of Hg+LED lamp? More details can be given.

Answer:

This publication is part of the research work of the doctoral student. In the conducted research series, the same variants cured on a paint line using mercury lamps and mercury-doped gallium were also considered. In this article, we allowed ourselves to refer to the obtained results of the resistance of coatings cured using different technologies (article titled "Mechanical and Chemical Resistance of UV Coating Systems Prepared under Industrial Conditions"), indicating the possibility of using LED lamps for curing coatings in the furniture industry. The next publication will be a summary and discussion of differences (mechanical and chemical resistance, as well as optical properties of coatings such as color and gloss) between traditional curing methods (mercury lamps and mercury-doped gallium) and the use of LED lamp modules and mercury lamps.

Reviewer 2 Report

Comments and Suggestions for Authors

The article was written to show how certain technical and procedural factors in final production affect the durability of coatings on furniture elements. The presented literature results and the authors' studies are intended to show that Hg-Ga lamps can be replaced by Hg+LED modules in industrial final production.

The manuscript has the character of an overview article, interspersed with statistical test evaluations. It presents an interesting parametric study on the mechanical and chemical resistance of coating systems on HDF boards. The special feature of the investigations is that the coatings were produced under industrial conditions.

Abstract

The statement formulated in the abstract: “The findings confirm the conception of that Hg+LED lamp modules can replace Hg-Ga lamps.” is not proven in the work. Experimental data allowing comparison between the two curing methods are not provided but should be added.

Materials and Methods

Information on the composition of the base and top coat is missing here. Depending on the material, the conditions under which a paint hardens (wavelength/energy of the radiation at which the reaction begins) differ. The reader can only tell that these are acrylic varnishes from Figure 1.

Which acrylic material was used as the base coat and which was used as the top coat? What components do the coatings consist of? Add this important information.

The chemical composition of the coating materials is relevant to the curing conditions and must be specified. This fact is not taken into account at all.

Under what conditions do these materials cure (wavelength range, surface power density of the UV source)? Why was a 395 nm LED lamp used and not a 405 nm LED lamp? Compare it to the emission spectrum of a Hg-Ga lamp.

Throughout the article, the authors talk about the power of the mercury and LED lamps used and indicate this in the unit W/cm. Firstly, the correct unit is W/cm2. Secondly, this is a surface power density. This quantity is used to express the intensity of electromagnetic radiation. The incorrect name and unit of measurement are repeated throughout the article, including the figures, and need to be corrected.

Results

Check the equation on line 282. The unit may not be correct. The unit of elastic modulus is GPa.

Conclusions

In the conclusions the authors refer to other publications by the authors. Where is the data supporting the improved chemical and abrasion resistance of Hg+LED cured coatings compared to Hg+Ga lamps.

The next sentence describes the opposite: “Coatings cured with LED+Hg lamps achieved lower abrasion resistance…” Check the statement.

The abbreviation Hg+Ga lamps used here is incorrect. Mercury and gallium-doped mercury lamps were probably used.

Lines 136, 137: The sentence “The LED lamp was an air-cooled light array with a peak range of 12W/cm, centered at 395 nm.” is misleading.

Check the spelling on lines 148 and 200.

Check the spelling of the chemical compound on line 381.

Author Response

Dear Reviewer,

thank you very much for your valuable comments. Our answers are listed below and in the file.

yours sincerely,

Tomasz Krystofiak

Comments and Suggestions for Authors

The article was written to show how certain technical and procedural factors in final production affect the durability of coatings on furniture elements. The presented literature results and the authors' studies are intended to show that Hg-Ga lamps can be replaced by Hg+LED modules in industrial final production.

The manuscript has the character of an overview article, interspersed with statistical test evaluations. It presents an interesting parametric study on the mechanical and chemical resistance of coating systems on HDF boards. The special feature of the investigations is that the coatings were produced under industrial conditions.

Answer:

Thank you very much for your valuable comments.

Abstract

The statement formulated in the abstract: “The findings confirm the conception of that Hg+LED lamp modules can replace Hg-Ga lamps.” is not proven in the work. Experimental data allowing comparison between the two curing methods are not provided but should be added.

Answer:

This work presents the results of testing the mechanical resistance of coatings hardened using LED lamps. It is a continuation of the authors' previous work ("Mechanical and Chemical Resistance of UV Coating Systems Prepared under Industrial Conditions"), which examined similar application variants cured on a painting line using mercury and gallium-doped mercury lamps. An initial comparison of the resistance outcomes of coatings hardened using various methods is shown in this article, suggesting that LED lamps may be used to harden coatings in the furniture sector. The publications are part of the PhD student's research work. The next publication will provide an overview and analysis of the differences between hardening coatings using LED and mercury lamp modules and traditional methods (mercury and gallium-doped mercury lamps) as well as the mechanical and chemical resistance and optical properties of coatings like color and gloss. Sentence used in the abstract: “The findings confirm the conception of that Hg+LED lamp modules can replace Hg-Ga lamps.” - it was used too early at this stage. Has been deleted.

Materials and Methods

Information on the composition of the base and top coat is missing here. Depending on the material, the conditions under which a paint hardens (wavelength/energy of the radiation at which the reaction begins) differ. The reader can only tell that these are acrylic varnishes from Figure 1.

Which acrylic material was used as the base coat and which was used as the top coat? What components do the coatings consist of? Add this important information.

The chemical composition of the coating materials is relevant to the curing conditions and must be specified. This fact is not taken into account at all.

Answer:

The primary objective of the study was to examine the impact of specific factors on the finishing line affect the resistance of coatings on furniture elements. The work that is being presented focuses on the influence of technological factors, not on the chemical composition. We agree that composition plays a significant role in UV curing procedures. To guarantee that the polymerization reaction proceeds as intended, it is crucial to maintain the proper ratios between each component and to use the appropriate amount. Standard commercial products were used for this investigation instead of developing any formulations for varnish products. Their ingredients were conditionally approved for publication (without specifying the company brand) following conversation with the manufacturer. However, their content is confidential information. Therefore, detailed data cannot be shared.

Under what conditions do these materials cure (wavelength range, surface power density of the UV source)? Why was a 395 nm LED lamp used and not a 405 nm LED lamp? Compare it to the emission spectrum of a Hg-Ga lamp.

Answer:

To effectively apply UV LED technology, it is necessary to adapt the entire production process. On an industrial technological line, the experiments were run. As part of its activities related to the varnishing process of wood-based boards, the company purchased a technological line equipped with LED lamp modules with a wavelength of 395 nm. The line and painting technology were developed in a project implemented with the support of the National Center for Research and Development in Poland. Scientists working with the company particularly chose paint supplies to meet the requirements of the prototype technological line.

Throughout the article, the authors talk about the power of the mercury and LED lamps used and indicate this in the unit W/cm. Firstly, the correct unit is W/cm2. Secondly, this is a surface power density. This quantity is used to express the intensity of electromagnetic radiation. The incorrect name and unit of measurement are repeated throughout the article, including the figures, and need to be corrected.

Answer:

Thank You for pointing out the mistake, we have fixed the mistake in the text and charts.

Results

Check the equation on line 282. The unit may not be correct. The unit of elastic modulus is GPa.

Answer:

The data has been verified

Conclusions

In the conclusions the authors refer to other publications by the authors. Where is the data supporting the improved chemical and abrasion resistance of Hg+LED cured coatings compared to Hg+Ga lamps.

The next sentence describes the opposite: “Coatings cured with LED+Hg lamps achieved lower abrasion resistance…” Check the statement.

Answer:

Thank You for pointing out the mistake. . Certainly, curing coatings using LED modules and mercury lamps has improved chemical resistance and scratch resistance. The word 'abrasion' was incorrectly used; the correct term is 'scratch’. The following sentence is correct: “Coatings cured with LED+Hg lamps achieved lower abrasion resistance…”

 The abbreviation Hg+Ga lamps used here is incorrect. Mercury and gallium-doped mercury lamps were probably used.

Answer:

Thank You for pointing out the mistake, we have fixed the mistake in the text and charts.

Lines 136, 137: The sentence “The LED lamp was an air-cooled light array with a peak range of 12W/cm, centered at 395 nm.” is misleading.

Answer:

The sentence was deleted and the information about the length of the lamps was moved higher.

Check the spelling on lines 148 and 200.

Check the spelling of the chemical compound on line 381.

Answer:

The data has been verified.
